# HDL-like-Mediated Cell Cholesterol Trafficking in the Central Nervous System and Alzheimer’s Disease Pathogenesis

**DOI:** 10.3390/ijms23169356

**Published:** 2022-08-19

**Authors:** Carla Borràs, Aina Mercer, Sònia Sirisi, Daniel Alcolea, Joan Carles Escolà-Gil, Francisco Blanco-Vaca, Mireia Tondo

**Affiliations:** 1Institut d’Investigació Biomèdica Sant Pau (IIB), Sant Quintí 77-79, 08041 Barcelona, Spain; 2CIBERDEM, ISCIII, 28029 Madrid, Spain; 3Department of Biochemistry and Molecular Biology, Universitat Autònoma de Barcelona, 08193 Bellaterra, Spain; 4Sant Pau Memory Unit, Department of Neurology, Hospital de la Santa Creu i Sant Pau, 08041 Barcelona, Spain; 5CIBERNED, ISCIII, 28029 Madrid, Spain; 6Department of Biochemistry, Hospital de la Santa Creu i Sant Pau, 08041 Barcelona, Spain

**Keywords:** Alzheimer’s disease, apolipoprotein E, cholesterol trafficking, HDL, cholesterol efflux, central nervous system, dementia

## Abstract

The main aim of this work is to review the mechanisms via which high-density lipoprotein (HDL)-mediated cholesterol trafficking through the central nervous system (CNS) occurs in the context of Alzheimer’s disease (AD). Alzheimer’s disease is characterized by the accumulation of extracellular amyloid beta (Aβ) and abnormally hyperphosphorylated intracellular tau filaments in neurons. Cholesterol metabolism has been extensively implicated in the pathogenesis of AD through biological, epidemiological, and genetic studies, with the *APOE* gene being the most reproducible genetic risk factor for the development of AD. This manuscript explores how HDL-mediated cholesterol is transported in the CNS, with a special emphasis on its relationship to Aβ peptide accumulation and apolipoprotein E (ApoE)-mediated cholesterol transport. Indeed, we reviewed all existing works exploring HDL-like-mediated cholesterol efflux and cholesterol uptake in the context of AD pathogenesis. Existing data seem to point in the direction of decreased cholesterol efflux and the impaired entry of cholesterol into neurons among patients with AD, which could be related to impaired Aβ clearance and tau protein accumulation. However, most of the reviewed studies have been performed in cells that are not physiologically relevant for CNS pathology, representing a major flaw in this field. The ApoE4 genotype seems to be a disruptive element in HDL-like-mediated cholesterol transport through the brain. Overall, further investigations are needed to clarify the role of cholesterol trafficking in AD pathogenesis.

## 1. Introduction

### 1.1. Alzheimer’s Disease

The World Human Organization estimates that 135 million people will have dementia by the year 2050 [1]. Among dementia cases, the most common form is Alzheimer’s disease (AD), a progressive and devastating neurodegenerative disorder, usually related to aging, that represents 60–80% of all dementia cases [2]. Alzheimer’s disease is a disorder that triggers difficulty in communicating and reasoning; mood changes; progressive memory loss; and, in due course, the loss of independent living. Histologically, AD is characterized by the pathologic accumulation of extracellular amyloid beta (Aβ) and abnormally hyperphosphorylated intracellular tau filaments in neurons, leading to senile plaques and neurofibrillary tangles, respectively, with neuropathological lesions preceding clinical signs by many years [3,4,5]. Currently, there is still no successful therapeutic strategy for disease mitigation.

### 1.2. Involvement of Lipids in AD Pathology

The majority of the brain is composed of lipids, which can be grouped into three main categories, sphingolipids, glycerophospholipids, and cholesterol [6]. Lipids have been associated with a healthy brain, participating in the function of the blood–brain barrier (BBB), the processing of amyloid precursor protein (APP) in lipid rafts, myelination, membrane remodeling, receptor signaling, oxidation, inflammation, and energy balance [7]. Specifically, cholesterol plays an important role during the developmental stage and in adult life in terms of the overall maintenance of brain health, including neuron repair, membrane remodeling, and plasticity [8]. The brain is considered a cholesterol-rich organ because it contains about 25% of the entire body’s cholesterol [9], which is present in the membranes of neurons, glial cells, and myelin membranes [10]. Due to the incapacity of cholesterol to traverse the BBB, the central nervous system (CNS) depends almost exclusively on de novo endogenous synthesis, which, in adults, is mostly performed by astrocytes [11].

Cholesterol metabolism has been extensively implicated in the pathogenesis of AD through biological, epidemiological, and genetic studies [12,13,14,15,16,17,18,19,20,21]. In this sense, genome-wide association studies have identified several cholesterol-metabolism-related genes as top risk factors for late-onset AD. The strongest AD cholesterol metabolism susceptibility *loci* include genes such as *APOE*, *BIN1*, *CLU* (alias *APOJ*), *PICALM*, *ABCA7*, *ABCG1*, *SREBF2*, and *SORL1*, among others [7,12,15,22,23]. Nonetheless, only *APOE* has been considered significantly associated with amyloid or tangle pathologies in AD [24]. In this sense and as discussed below, the *E4* allele of the *APOE* gene encoding apolipoprotein E (ApoE) was described, more than three decades ago, as the most robust and reproducible genetic risk factor for the development of AD [18,25]. Other lipids such as fatty acids, glycerolipids, glycerophospholipids, and sterols and sphingolipids such as ceramides and sphingomyelin have also been involved in AD pathogenesis. For instance, ceramides promote the production and accumulation of Aβ as they stabilize the beta-site APP-cleaving enzyme, and sphingomyelins have a binding motif to Aβ, displaying a role in its aggregation. Furthermore, its imbalance has also been shown to contribute to AD development [7,26]. 

In the periphery, high-density lipoprotein (HDL) particles are the main physiological acceptors of cellular cholesterol from all extrahepatic body compartments, including the intimal macrophage foam cells of atherosclerotic lesions [27]. Cholesterol efflux initiates the reverse cholesterol transport pathway, which conveys cholesterol from peripheral cells to the liver for its fecal excretion, representing a major anti-atherosclerotic pathway in the organism [28]. Regarding the CNS, HDL cholesterol trafficking seems to undergo processes similar to those observed for plasma HDL, with significant modifications. 

In the present review, we aim to describe the mechanisms by which HDL-like trafficking through the CNS occurs, with special emphasis on HDL-like-mediated cholesterol efflux and uptake processes and their potential implications in AD pathogenesis. 

### 1.3. Literature Search Strategy

A literature review was performed based on the “Preferred Reporting Items for Systematic Reviews and MetaAnalyses” (PRISMA) statement. Relevant studies from peer-reviewed journals were identified from three electronic databases (PubMed, Google Scholar, and Web of Science) up to 1 June 2022, without any language restriction. Four groups of medical subject terms were applied, including “Alzheimer disease”, “cholesterol trafficking”, “central nervous system”, and “cholesterol efflux”. To identify additional studies and reviews, combinations of specific keywords were also performed: dementia, ApoE, and HDL. The hand searching of reference lists in the included reviews was also performed. Three of the authors (C.B., J.C.E-G. and M.T.) independently screened articles, extracted relevant data, and assessed the quality of the studies. For the works exploring cholesterol efflux and cholesterol uptake processes regarding AD in different cell types, a uniform table was prepared to collect related characteristics, including the first author, year of publication, cell culture, sample used (cholesterol acceptor), mechanism tested, activation, and main findings. All papers describing findings related to the mechanisms underlying HDL-like-mediated cholesterol trafficking in the CNS and its implications in AD pathogenesis were included, whereas similar papers related to non-AD dementia were excluded. The process was agreed upon by all authors.

## 2. HDL-Mediated Cholesterol Trafficking in the CNS

### 2.1. Transporters of Cholesterol in the Brain

Cholesterol trafficking in the CNS involves several transporters with similar roles to those in the peripheral cells [29], reinforcing the relevance of the control of lipid homeostasis in the brain. Identically to peripheral tissues, cholesterol efflux in the brain takes place through aqueous diffusion facilitated by scavenger receptor class B type I (SR-BI) and by active pathways involving the ATP-binding cassette (ABC) transporters A1 and G1 (ABCA1 and ABCG1) [30]. In contrast to the periphery, where SR-BI is ubiquitous, in the brain, it is only expressed in astrocytes, neurons, and capillary endothelial cells [31,32], as it is regulated by the sterol regulatory element-binding protein 2 (SREBP-2) transcription factor binding sites [33,34]. In the CNS, ABC transporters have been found in neurons, astrocytes, and capillary endothelial cells, participating in protein secretion and lipidation [35,36]. Specifically, ABCA1 acts as the main cholesterol efflux regulatory protein, interacting with mildly lipidated ApoE particles to form small HDL-like lipoproteins containing phospholipids (PL) and unesterified cholesterol (UC) [29]. SR-BI and other transporters from the ABC family, such as ABCG1 and ABCG4, interact with already-lipidated forms [37], thereby completing the lipidation of small HDL-like particles and generating larger ones via the addition of more cholesterol as well as other lipids [38]. Once in the extracellular space, lipoproteins can be further enriched with ApoE [39,40,41].

ABCA1, ABCG1, and ApoE gene transcription can be modulated by the liver X receptors (LXRs) α and β, ligand-activated transcription factors that bind to DNA and form heterodimers with retinoid X receptors (RXRs) to exert their functions [42,43]. Oxysterols and 9-cis-retinoic acid (RA) are, in turn, endogenous ligands for LXRs and RXRs, respectively [44,45]. Thus, a high concentration of intracellular cholesterol or its derivatives in astrocytes activates LXR/RXR-mediated transcription for cholesterol transport proteins to facilitate efflux [46,47]. In this sense, the co-expression of ApoE with ABCA1 by the LXR/RXR system reinforces the important role of ApoE lipidation in the efflux process [48]. ABCA1 may also play a critical role in removing excess cholesterol from neurons, which can either be converted into cholesterol esters and kept in the cytoplasm or converted to 24S-hydroxycholesterol (HC) by 24-hydroxylase [49,50], as discussed below. 24-HC may reach the astrocytes and, through the activation of LXR, inhibit cholesterol synthesis and upregulate ABCA1, ABCG1, and ApoE levels [44]. Alternatively, it can be exported from the brain through the BBB [41,51,52,53]. Concerning ABCA7, this transporter is highly expressed in the brain, sharing significant homology with ABCA1 [35]. However, its transcription is downregulated when intracellular cholesterol concentrations are high [51] and upregulated through the SREBP-2 pathway when they are low [54]. Its role in cholesterol efflux is less known and seems to be less relevant than those of the other ABC proteins [55].

In addition to endogenous LXR/RXR agonists, exogenous ligands have also been described. T0901317 is a synthetic and highly selective agonist for LXRs, with a demonstrated enhancing effect on the ABCA1 and ABCG1 transporters and cholesterol efflux [42]. Similarly, ABCA1 activity can also be modulated by cyclic adenosine monophosphate (cAMP) via a protein kinase A (PKA)-dependent pathway. Protein kinase A increases *ABCA1* gene transcription and phosphorylates ABCA1 protein, increasing its ability to export cholesterol [56].

Regarding cholesterol uptake, mature HDL-like particles deliver cholesterol to brain cells through the interaction of ApoE with specific lipoprotein receptors [57], including the low-density lipoprotein (LDL) receptor (LDLR), LDL-receptor-related protein 1 (LRP1), the very low density lipoprotein (VLDL) receptor (VLDLR), and the ApoE receptor (ApoER2), which is mainly expressed in the brain [58]. All of them bind ApoE and lipidated ApoE with different degrees of affinity [29,59]. Characteristically, nascent lipoproteins secreted by astrocytes show a higher affinity for LDLR, whereas CSF ApoE-containing HDL-like particles adhere strongly to LRP1 [60]. Despite being present in both neurons and astrocytes, LDLR is most highly expressed in glia, whereas LRP1 is more highly expressed in neurons [61,62,63,64,65]. Specifically, LRP1 is a critical source of cholesterol for neurite outgrowth, synaptogenesis, and remodeling; however, it may also exert negative feedback to limit the intracellular cholesterol concentration [66,67]. Regarding VLDLR and ApoER2, they are structurally very similar to the LDLR; however, they mainly bind other ligands involved in neurodevelopment and synaptic functions, such as reelin [68]. The information regarding cholesterol transporters and receptors potentially involved in cholesterol processes in the brain is summarized in Table 1.

With respect to lipid movement through the central and peripheral compartments, the BBB and the blood–cerebrospinal fluid barrier (BCSFB) act as semipermeable membranes, regulating the exchange of solutes between the blood and CNS [69]. Therefore, all lipoproteins, including LDL, VLDL, and chylomicrons, are excluded from the brain [70]. ApoA-I’s presence in the CSF derives from the blood circulating HDL [71,72]. The ABCA1 and ABCG1 transporters present on epithelial cells of the BBB can mediate the lipidation of peripheral apolipoproteins after their entry into the CNS [48,73], facilitating the formation of ApoE/ApoA-I small HDL-like particles [34]. Moreover, small HDL plasma particles can enter the brain via SR-BI-mediated uptake and transcytosis [34].

In contrast to cholesterol, the oxysterols 24-HC and 27-hydroxycholesterol (27-HC) (oxidized cholesterol metabolites) can cross the BBB and BCSFB at no energetic cost [74]. In that sense, 27-HC is synthesized by CYP27A1, representing the major cholesterol metabolite in the circulation. Its expression takes place in most of the organs and tissues [75] and can be transported by diffusion from the circulation into the brain [76]. Recent works support the idea that the pool of 27-HC contributes to cholesterol-related metabolite uptake in neurons [77]. On the other hand, due to the limited capacity of neurons and glial cell to eliminate cholesterol, spare cholesterol must reach the liver for further conversion to bile acids and final excretion. As previously stated, this function can rely on 24-HC, the main available hydrophilic form of cholesterol in the brain, to be transferred through the BBB [42,51,52,53]. 24-HC is originally released by neurons through the neuron-specific enzyme CYP46A1 [53]. A secondary excretion pathway through the BBB may involve cholesterol efflux mediated by ABCA1 and ABCG1 [34,78]. The steady-state levels of both oxysterols are tightly regulated in the brain. Therefore, disturbances in their concentrations have been associated with different forms of dementia [79]. In that sense, several works have evaluated plasma or CSF oxysterol levels in patients with AD, with evidence of impaired concentrations [80,81,82]. Particularly, 24-OHC has been found to regulate APP via the production of the amyloidogenic fragment [83], whereas 27-HC may contribute to amyloid deposition [74]. Higher levels of the metabolites 24-HC and 27-HC in the CSF suggest an increase in the cerebral cholesterol load, as observed in AD brains in postmortem examinations [84]. Overall, these results suggest a key role of these cholesterol metabolites in AD pathology. However, the relationship of HDL-like lipoproteins with these metabolites in the CNS remains largely unknown.

### 2.2. Apolipoprotein E in the CNS in the Context of AD

Excellent reviews focusing almost exclusively on ApoE and AD exist [85,86,87,88,89,90,91,92,93,94]. As previously stated, liver-synthesized ApoE cannot cross the BBB [95]. Thus, the ApoE in the CNS is synthesized by astrocytes and microglia [96]. In stressful situations, neurons can also produce ApoE [97]. It is worth noting that despite the fact that some authors refer to ApoE as a relevant AD component on its own, it is indeed a relevant part of the HDL-like lipoprotein structure that transports multiple proteins and lipid species, thereby affecting their CNS metabolism. Accordingly, ApoE is the main lipid carrier in the CNS, with functions that include the transport of lipids (mainly cholesterol) between neurons and glial cells via interactions with transporters such as ABCA1 and ABCG1 [98], the regulation of lipid metabolism [99], and the enhancement of axonal growth through LDLR interactions [60]. ApoE is also essential for the brain homeostasis-regulating processes of Aβ clearance as well as for the inhibition of inflammatory pathways, both functions of great importance in the AD brain, where Aβ oligomers and neuroinflammatory metabolites tend to accumulate [100]. Specifically, lipidated ApoE clears Aβ peptides from the brain [101]. In that sense, it is a well-established fact that the ApoE lipidation degree is positively associated with its affinity for soluble Aβ, with poorly lipidated ApoE obstructing Aβ clearance and stimulating Aβ deposition [25,102,103,104], thus representing a risk factor for AD, as developed below.

#### 2.2.1. Aβ Peptides, Cholesterol Transporters, and ApoE Interplay

Extracellular Aβ clearance occurs through different mechanisms. In that regard, the interplay between ApoE, Aβ peptides, transporters, and receptors is crucial. In brief, Aβ clearance involves three main mechanisms: glial phagocytosis, protease degradation, and transport to the periphery through the BBB/BCSFB or to cervical lymph nodes [105,106]. Glial phagocytosis consists of the phagocytosis of Aβ by astrocytes and microglia through ABCA7 [107], whereas proteolytic degradation involves enzymes such as neprilysin, insulin-degrading enzyme, and endothelin-converting enzyme [106]. Ultimately, Aβ transport to the periphery allows Aβ clearance by blood components or even by tissues and organs such as the liver and kidneys [105]. Specifically, transport through the BBB and BCSFB involves the LRP1, ABCB1, ABCG4, ABCA7, and VLDLR transporters [106,108,109].

Beyond these mechanisms, some studies demonstrated that ABCA1 may positively influence Aβ clearance [43,45], despite Aβ not directly binding to this receptor. As such, a recent work demonstrated that the greater propensity of ApoE4 to aggregate decreased ABCA1 membrane recycling and its ability to lipidate ApoE, thus displaying a negative regulation loop that may enhance AD progression [110]. Significantly, enhancing ABCA1 activity to lipidate ApoE reduced ApoE and ABCA1 aggregation [110]. Similarly, another work demonstrated that a biomimetic HDL was able to cross the BBB in vitro and compensate for reduced levels of ABCA1 due to Aβ-induced astrogliosis, thus promoting cholesterol efflux from astrocytes [111].

#### 2.2.2. ApoE4 and AD Pathogenesis

ApoE is a protein composed of 299 amino acids divided into three domains: the N-terminal domain (residues 1–191), responsible for binding to the LDLR family; the C-terminal domain (residues 206–209), containing the lipid-binding region; and the unstructured hinge regions that allow protein mobility [112]. In humans, ApoE can be found in three isoforms, E2, E3, and E4, all with different molecular and functional properties and determined by three different alleles at a single genetic locus. The three isoforms differ by a change in one or two amino acids at residues 112 or 158 [113]. The importance of these isoforms in modulating the pathogenesis of AD is a well-known fact. In this sense, ApoE3 is the most common isoform in humans, accounting for around 80% in some ethnic groups, but it is without any effect on AD predisposition [114]. In contrast, the relatively rare ApoE2 is considered neuroprotective [29,115], whereas ApoE4 is considered the strongest genetic risk factor for sporadic late-onset AD [97,116,117]. In this regard, ApoE4 demonstrates a lower affinity for lipids, limiting their transport through the CNS, which is needed for repair and neuronal remodeling [118]. Additionally, ApoE4 carriers are more susceptible to oxidative stress and lipid peroxidation [112].

The involvement of ApoE4 isoforms in AD pathogenesis occurs through various mechanisms, including, among others, effects on Aβ metabolism, tau protein, and lipid metabolism regulation [86], as summarized in Table 2. These pathogenic mechanisms add to additional ApoE effects, including oxidative stress and neuroinflammation [119]. The main effects of the ApoE4 isoform on Aβ metabolism include the impairment of the following processes: altered Aβ production by the beta-site APP-cleaving enzyme 1 (BACE-1), altered ApoE binding, altered clearance, and altered aggregation and deposition. Indeed, when compared to ApoE2 and ApoE3, ApoE4 was described to enhance Aβ production and BACE-1 levels [120,121], to increase Aβ binding affinity to ApoE [25,103], to decrease Aβ clearance [122,123,124,125], and to facilitate Aβ aggregation and deposition [126]. Consequently, ApoE4 negatively affects Aβ metabolism, thus contributing to AD progression. Concerning tau pathology, the ApoE genotype may also affect tau protein, despite not directly interacting with it. Specifically, ApoE4 was reported to worsen neurodegeneration in human primary tauopathies [127], which could be aggravated when amyloid pathology was also present [24]. A recent study also investigated the mechanisms by which ApoE4 expression in neurons increased tau phosphorylation and enhanced the release of phosphorylated tau. In particular, ApoE4 predisposed neurons to accelerated neurodegeneration through a heparin sulfate proteoglycan-dependent mechanism [128]. Of note, LRP1 was recently shown to be a master receptor for tau uptake by neurons [129]. In that sense, ApoE4 could affect the ability of tau to bind LRP1. The study by Rauch et al. demonstrated that all ApoE isoforms reduced tau uptake in vitro to a similar extent. However, due to its low circulating levels, ApoE4 would have low efficiency for inhibiting tau–LRP at physiologically relevant levels in vivo [129]. Finally, regarding the regulation of lipid metabolism, ApoE4 astrocytes have been reported to present with a higher expression of genes involved in cholesterol biosynthesis, therefore displaying lipid metabolic dysregulation and cholesterol accumulation [130,131]. Whether these alterations affect HDL-like synthesis, remodeling, and cholesterol transport between CNS cells remains unknown.

### 2.3. HDL-like Lipoprotein Metabolism in the CNS

To understand the HDL-mediated cholesterol efflux and uptake processes through CNS cells in AD, a brief explanation of the biogenesis, remodeling, and delivery of HDL-like particles is provided below.

Like plasma, the lipoproteins present in the CSF are composed of cholesterol, phospholipids, and apolipoproteins. Overall, their brain concentration is very low, representing approximately 1–10% of their plasma levels [132]. Brain lipoproteins present with a similar size and density to peripheral HDL particles, and therefore they are defined as “HDL-like particles” [132]. However, brain lipoproteins present with unique traits, including a wider size range (8–22 nm) and a different apolipoprotein composition, with ApoE being the major protein component [99]. In addition to ApoE, glial-derived discoidal HDL has ApoJ, whereas mature spherical CSF HDL has small amounts of ApoA-I. As stated above, ApoA-I is the most important constituent of plasma HDL; however, it is not produced in the brain and is, rather, delivered to the CNS through the BBB [133]. Other minoritarian apolipoproteins present in the brain include ApoA-II, ApoA-IV, ApoD, and ApoH [132]. Despite not being fully elucidated, the mechanisms involved in CSF lipoprotein synthesis and remodeling are similar to those observed for plasma HDL. It is well-known that astrocytes and microglia are responsible for the synthesis of most lipoproteins found in the brain and CSF [134]. Remodeling enzymes and lipid transfer proteins have also been identified in the CNS [135]. Lecithin-cholesterol acyltransferase (LCAT) converts UC and phosphatidylcholine to cholesteryl esters [136] via ApoE activation [137]. In fact, LCAT can be synthesized in the liver and testes as well as in astrocytes [137], suggesting its important role in the remodeling and maturation of nascent HDL-like proteins into larger spherical particles [99]. It has been found at concentrations representing around 5% of the levels in plasma [99,138]. Other remodeling enzymes include phospholipid transfer protein (PLTP), which is reported to be at a concentration representing about 15% of the plasma level [139,140], and cholesteryl ester transfer protein (CETP), which is reported to be present at about 12% of the level found in plasma [140] or non-existent [141]. Finally, as described above, cholesterol can be delivered to neurons by mature HDL-like lipoproteins through interactions with specific receptors [8]. A detailed description of HDL-like-mediated cholesterol trafficking in healthy and AD brains is shown in Figure 1.

### 2.4. Cholesterol Efflux to HDL-like Particles in AD

Brain cholesterol efflux is a key step in cholesterol trafficking through the CNS. As previously described, the process consists of the transference of intracellular cholesterol to extracellular acceptors, including apolipoproteins and HDL-like lipoproteins, through aqueous diffusion or SR-BI and ABC-family transporters.

The first study that aimed to describe the ability of brain lipoproteins to promote cholesterol efflux was performed in 1998 in fibroblasts [142]. This study was pioneering in demonstrating the critical role of CSF lipoproteins in brain cholesterol trafficking, thus opening the door for further studies. Since then, researchers have explored the ability of cholesterol acceptors, including serum, serum-isolated, or recombinant apolipoproteins, lipoprotein, and CSF, to promote cholesterol efflux in different cell types. All published studies regarding cholesterol trafficking in the brain are summarized in Table 3. Most of these works focused on the molecular pathways by which cholesterol efflux takes place, proving that the ABCA1 and ABCG1 transporters are pivotal in this process [39,41,45,52,143,144]. Specifically, the work of Koldamova et al. explored the role of ABCA1 in cholesterol efflux to acceptors such as serum, ApoA-I, and recombinant ApoE3 in primary rat neuron culture, astrocytes, and other glial cells. The authors found that, after exposure to 22(R)-HC and 9-cis-RA, ABCA1 expression and protein levels as well as ApoA-I and ApoE-mediated cholesterol efflux increased [45]. In line with these findings, the cholesterol efflux to these acceptors was reduced in ABCA1-deficient astrocytes and microglia when compared to wild-type cultures [39]. Subsequent works also showed that the stimulation of ABCA1 and ABCG1 with cAMP or 22(R)-HC plus 9-cis-RA was able to enhance cholesterol efflux in astrocytes [52,143]. Another work showed that Abca1-deficient mice presented with small APOE-containing lipoproteins. Moreover, their cultured astrocytes secreted lipoproteins that presented with markedly reduced cholesterol concentrations [41]. Similarly, primary cultured astrocytes from *Abcg1* and *Abcg4* knockout mice displayed defective cholesterol efflux to HDL-like particles, while the overexpression of these transporters in hamster embryonic kidney (HEK)-293 cells by transfection resulted in increased cholesterol efflux [144]. Regarding ABCG4, a study in primary cortical rat astrocytes reported that the inhibition of ABCG4 did not affect cholesterol efflux. However, the authors also showed that ABCG4 expression was crucial in mediating this process in primary neurons [52].

Considering the effect of ApoE isoforms on AD pathology, some works evaluated how they affect efflux capacity and obtained divergent results. While some authors found similar rates of cholesterol efflux from HDL-like particles in neuronal cells and astrocytes regardless of the ApoE isoform [145,146], others demonstrated that recombinant ApoE3 was able to induce a higher lipid efflux compared to recombinant ApoE4 in neuronal and astrocytic cell cultures [147]. Accordingly, another work also found that primary astrocytes from modified mice expressing human ApoE4 displayed lower cholesterol efflux than those cells from mice expressing human ApoE3 [110]. Interestingly, the accumulation of fat in astrocytes, a stress-associated condition, induced the assembly and secretion of potentially toxic triacylglycerol-rich ApoE-containing lipoproteins, a process enhanced by ApoE4 [146]. On the whole, the contribution of the ApoE genotype to cholesterol efflux and its pathophysiological consequences requires further studies. Finally, another line of work has focused on exploring the cholesterol efflux capacity of human samples taken from control individuals and AD patients. Several studies using cultured murine macrophages found reduced cholesterol efflux in AD patients’ samples when using serum as an acceptor [30,148]. Despite these interesting results, some authors argued that CSF is the most suitable physiological sample to study cholesterol trafficking in the brain because it contains all the components involved in central cholesterol efflux. Accordingly, the first approach assessing cholesterol efflux capacity in a small number of control and AD patient CSF samples in rat astrocytes found no differences between the two groups [141]. More recently, the evaluation of cholesterol efflux to the CSF in larger cohorts was addressed. Using baby hamster kidney (BHK) cells overexpressing ABCA1, the authors reported that CSF samples taken from AD patients displayed 30% less cholesterol efflux than CSF samples from cognitively healthy participants [149]. In agreement, a recent work performed with murine macrophages induced with cAMP to express ABCA1 at one end and Chinese hamster ovary (CHO)-K1 cells overexpressing ABCG1 at the other end resulted in decreased levels of cholesterol efflux from AD patients’ CSF compared to a control population [150]. Likewise, cholesterol efflux to the CSF from ApoE3 carriers was significantly increased compared to that from ApoE4 carriers in ABCA1-induced BHK cells [110]. Despite the importance of these findings, only one work has performed similar studies in cells that are functionally and morphologically relevant in the brain and, thus, for AD pathology [151], representing a proof of concept for CSF cholesterol efflux quantification in human neurological cells. Efflux measurement in neurons, microglia, and astrocytes allowed researchers to conclude that CSF cholesterol efflux levels are positively correlated with total cholesterol, ApoA-I, ApoE, and ApoJ concentrations. However, demographic and clinical data were unknown because the CSF samples were taken from anonymous patients, making it impossible to compare efflux among healthy controls and patients with AD [151].

### 2.5. Cholesterol HDL-like Uptake in AD

One of the first studies addressing cholesterol uptake in connection with AD was performed in 1998 [142]. Specifically, the work addressed the function of CSF lipoproteins with assays of cholesterol efflux and cholesterol uptake. The authors showed that CSF lipoproteins labeled with a fluorescent dye were internalized by neuroglioma cells and cultured primary neurons and astrocytes. Nonetheless, this field has scarcely been explored in later years because most studies focused almost exclusively on cholesterol efflux. A recent report demonstrated that astrocytic ApoE-derived lipoproteins, presumably HDL-like, can transport a variety of microRNAs that specifically downregulate the genes involved in neuronal cholesterol biosynthesis [152]. Importantly, these ApoE-induced effects were significantly abolished in neurons after silencing LDLR or LRP1. Furthermore, ApoE4 was less capable of regulating these neuronal pathways in this situation [152].

An important line of work regarding cholesterol uptake involves the proprotein convertase subtilisin/kexin type 9 (PCSK9), a protein whose main role is to regulate LDLR recycling. PCSK9 is present in the serum and CSF. In the periphery, PCSK9 degrades LDLR, while in neurons it degrades LDLR as well as other ApoE-binding receptors, including VLDLR, LRP1, and apoER2 [89], causing decreased cholesterol uptake. Interestingly, recent studies found that patients with AD presented with significantly increased PCSK9 concentrations [153,154], especially ApoE4 carrier patients [153]. These results involve cholesterol uptake as a potential pathological mechanism of AD. It should be noted that LRP1 is a critical receptor that mediates tau endocytosis and spread [129].

## 3. Lipid-Based Therapies in the CNS with Respect to AD

Many efforts have been focused on developing a treatment for AD. Considering the multifaceted form of the disease, numerous therapeutic targets have been explored over the years. Here, we summarize the most relevant lipid-based strategies that could potentially affect HDL-like cholesterol trafficking in the CNS.

Statin use represents the first-line of treatment for hypercholesterolemia and the prevention of cardiovascular diseases due to their lipid-lowering, antioxidant, and anti-inflammatory effects [155]. Statins are able to cross the BBB. Thus, their potential in preventing and treating AD was recently addressed [156]. In that sense, in vitro and in vivo studies reported neuroprotective effects by interfering with Aβ and tau metabolisms [157]. However, divergent results were obtained in human clinical trials [158]. Probucol, another lipid-lowering drug, was shown to increase CSF ApoE concentrations as well as reduce CSF tau and Aβ concentrations in a pilot trial in mild-to-moderate sporadic AD [159]. Currently, a phase II trial in 314 participants with mild-to-moderate AD is being performed [160]. However, the potential of these drugs to regulate CNS HDL-like cholesterol trafficking remains unknown.

An interesting strategy that has been also explored is the use of nuclear receptor agonists that induce the expression of ABC transporters and ApoE in the CNS, potentially improving HDL-mediated cholesterol trafficking. In that respect, the oral administration of bexarotene, an RXR agonist, enhanced the clearance of Aβ plaques and ameliorated cognitive functions in a mouse model of AD [161], although these results could not be reproduced in other studies [162]. Similarly, the combined LXR/PPAR (peroxisome proliferator-activated receptors) agonist treatment with GW3965 and pioglizatone reduced the soluble and deposited forms of Aβ in a mouse model of AD [163]. In line with these findings, treating an amyloid mouse model with the LXR agonist T0901317 improved Aβ clearance and cognitive deficits [164,165].

ApoE mimetic peptides have also been tested in AD animal models. The treatment with the mimetic CN-105 revealed decreased Aβ pathology and ameliorated memory deficits in a mouse model of AD [166]. Likewise, COG1410 exerted neuroprotective effects against tau pathology and neuroinflammation in a similar mouse model [167]. Using the same strategy, an intravenous treatment with human recombinant ApoA-I resulted in reduced levels of cerebral Aβ and neuroinflammatory markers in the AD mouse brain [168]. Comparably, the smaller and orally bioavailable ApoA-I mimetic peptide 4F inhibited Aβ deposition and displayed anti-inflammatory effects in different AD mouse models [169]. The function that these mimetic agents may exert in preclinical models include HDL remodeling, the promotion of cholesterol efflux, the sequestration of oxidized lipids, and the activation of anti-inflammatory processes [170]. However, it is still unclear whether all these actions could work in the human CNS.

Finally, considering that HDL-mediated cholesterol uptake in neurons could be altered in AD, ApoE receptors constitute potential treatment targets. Therefore, guiding future research on the potential benefits of cholesterol-lowering medications, such as PCSK9 inhibitors, may be an excellent target for the treatment of AD. However, in a large controlled clinical trial, the ApoE genotype did not significantly alter the relationship between the PCSK9 inhibitor evolocumab and cognitive decline [171].

Clearly, more research is needed to evaluate the effectiveness of these lipid-based strategies and their influence on CNS HDL-like cholesterol trafficking for AD treatment.

## 4. Concluding Remarks

Evidence seems to point in the direction of decreased cholesterol efflux to the CSF’s HDL-like lipoproteins among patients with AD. However, the studies addressing this issue in AD patients have been performed in cells that are not physiological relevant for CNS pathology. Following this definition, the potential of AD patients’ CSF to induce cholesterol efflux from human astrocytes has not yet been addressed in a well-defined cohort of AD patients, representing a major flaw for cholesterol efflux studies with regard to AD. Regarding HDL-like-mediated cholesterol uptake studies in AD, there are apparently indications that the entry of cholesterol into the neuron may be decreased, a fact that could be related to impaired Aβ clearance and tau protein accumulation in AD. Furthermore, the PCSK9-medited recycling of receptors involved in cholesterol uptake could be impaired in AD neurons. In that sense, the existing crosstalk between cholesterol uptake and tau proteins trough the LRP1 receptor seems to be another pathway worth exploring in connection with AD pathology. Finally, it is worth recalling that ApoE is the main structural apolipoprotein in HDL-like lipoproteins, with a high level of involvement in their CNS metabolism. Hence, another important line of research points to ApoE4 as a disruptive element in cholesterol transport through the brain because studies exploring its effect on efflux, uptake, and Aβ clearance show that ApoE4 impairment is stronger than that of other ApoE isoforms. Overall, further investigation is advisable to clarify the role that HDL-like-mediated cholesterol trafficking in the brain plays with reference to AD.

## Figures and Tables

**Figure 1 ijms-23-09356-f001:**
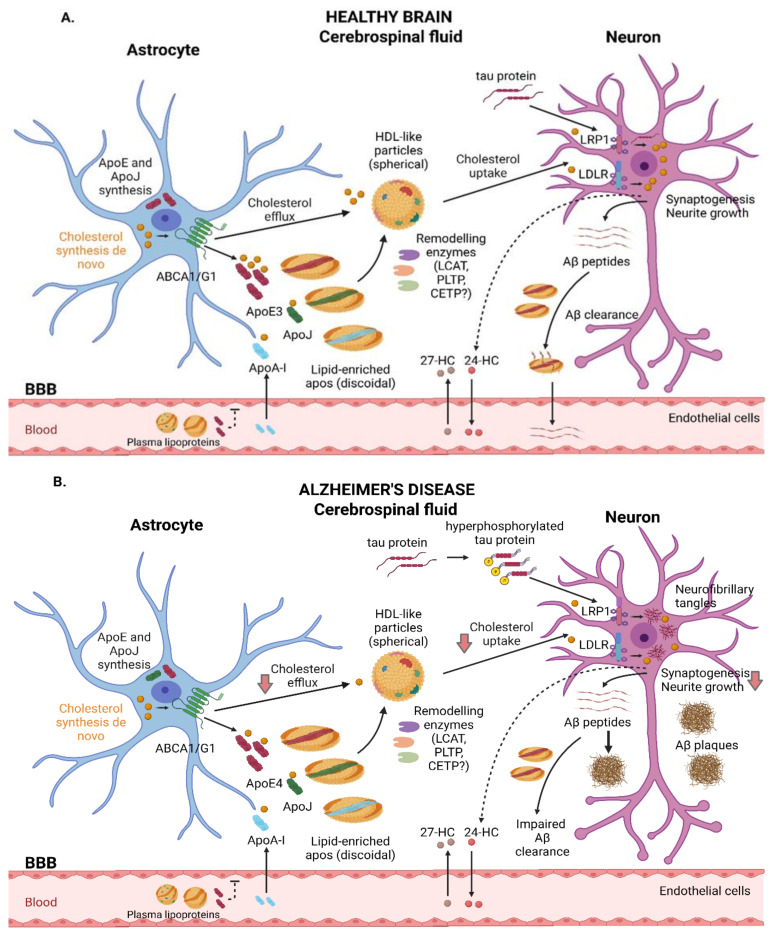
Schematic representation of the main steps involved in cholesterol trafficking in the brain. (**A**) In healthy subjects, astrocytes are responsible for de novo cholesterol and ApoE synthesis, with ApoE3 being the predominant isoform. Cholesterol efflux from astrocytes occurs, in part, through the ABCA1 and ABCG1 transporters. Lipid-free ApoE and, in smaller amounts, ApoA-I and ApoJ can be further lipidized by remodeling enzymes, resulting in spherical mature HDL-like particles that can interact with membrane receptors such as LRP1 and LDLR, leading to cholesterol uptake by neurons and guaranteeing essential functions such as synaptogenesis and neurite growth. Oxysterols can flux across the BBB. Neurons convert excess cholesterol in 24-HC, which can be eliminated to the bloodstream. In contrast, 27-HC enters the brain, where it promotes various functions. ApoE also contributes to the clearance of Aβ peptides. (**B**) In AD subjects, the pathological accumulation of hyperphosphorylated tau protein and Aβ plaque deposition may alter physiological functions in the brain. ApoE4, the predominant isoform in AD patients, is poorly lipidated and barely removes Aβ peptides. LRP1 plays a critical role in neuronal tau endocytosis. Recent works suggest alterations regarding cholesterol transport, including reduced HDL-like-mediated cholesterol efflux and impaired cholesterol uptake, leading to cell dysfunction. ABC: ATP-binding cassette; AD: Alzheimer’s disease; Apo: apolipoprotein; Aβ: amyloid beta; BBB: blood–brain barrier; CETP: cholesteryl ester transfer protein; HC: hydroxycholesterol; HDL: high-density lipoprotein; LCAT: lecithin-cholesterol acyltransferase; LDLR: low-density lipoprotein receptor; LRP1: LDLR-related protein 1.

**Table 1 ijms-23-09356-t001:** Cholesterol transporters and receptors potentially involved in cholesterol efflux and uptake processes in the brain.

Cholesterol Transporters and Receptors	Cellular Expression	Regulation	Main Functions
SR-BI	Astrocytes, neurons, and capillary endothelial cells	SREBP-2 pathway	Cholesterol diffusion to lipidated ApoE forms
ABCA1	Astrocytes, microglia, neurons, and capillary endothelial cells	LXR/RXR heterodimer/PKA-pathway	Cholesterol efflux to poorly lipidated ApoE
ABCG1	Astrocytes, neurons, and capillary endothelial cells	LXR/RXR heterodimer	Cholesterol efflux to lipidated ApoE forms
ABCG4	Astrocytes, microglia, neurons, and capillary endothelial cells	LXR/RXR heterodimer	Cholesterol efflux to lipidated ApoE forms
ABCA7	Astrocytes, neurons, and microglia	SREBP-2 pathway	Less known roles
LDLR	Astrocytes, microglia, neurons, and capillary endothelial cells	PCSK9	Cholesterol uptake regulator
LRP1	Astrocytes, microglia, neurons, and capillary endothelial cells	PCSK9	Cholesterol uptake regulator
VLDLR	Astrocytes, microglia, neurons, and capillary endothelial cells	PCSK9	Bind ligands for neurodevelopment and synaptic functions
ApoER2	Neurons	PCSK9	Bind other ligands involved in neurodevelopment and synaptic functions

ABC: ATP-binding cassette; Apo: apolipoprotein; ApoER2: apoE receptor 2; LDLR: low-density lipoprotein receptor; LRP1: LDL-receptor-related protein 1; LXR: liver X receptor; PCSK9: protein convertase subtilisin/kexin type 9; PKA: protein kinase A; RXR: retinoid X receptor; SR-BI: scavenger receptor class B type I; SREBP-2: sterol regulatory element-binding protein 2; VLDLR: very low density lipoprotein receptor.

**Table 2 ijms-23-09356-t002:** Deleterious effects of ApoE4 isoform in AD pathogenesis.

Effects of ApoE4 Genotype					
Aβ Metabolism		Tau Pathology		Lipid Metabolism	
↑ Aβ production	[120,121]	↑ Neurodegeneration	[127]	↑ Cholesterol synthesis and accumulation	[130,131]
↓ Aβ clearance	[122,123,124,125]	↑ Tau phosphorylation and secretion	[128]	↓ Lipid binding to ApoE	[118]
↑ Aβ binding to ApoE	[25,103]	↓ Tau binding to LRP1	[129]	↑ Oxidative stress and lipid peroxidation	[112]
↑ Aβ aggregation and deposition	[126]				

Apo: apolipoprotein; Aβ: amyloid beta.

**Table 3 ijms-23-09356-t003:** Brain cholesterol trafficking studies. The following table includes all works exploring cholesterol efflux and cholesterol uptake processes regarding AD in different cell types, ordered chronologically.

Cell Culture	Sample (Acceptor/Carrier)	Mechanism Tested	Activation	Main Findings	Reference
Fibroblasts	CSF lipoproteins	Baseline efflux	None	CSF lipoproteins induce cholesterol efflux	[142]
Neuroglioma cells and primary neurons and astrocytes	CSF lipoproteins	Uptake	None	CSF lipoproteins are internalized by neurons	
Rat astrocytes	CSF from AD (n = 3) and controls (n = 3)	Baseline efflux	None	No differences	[141]
Primary neurons	ApoA-I from human plasma and recombinant ApoE3	ABCA1-mediated efflux	25-HC, 9-cis-RA	25-HC, 9-cis-RA: ↑ efflux levels	[45]
Primary murine wild-type and Abca1−/− astrocytes and microglia	Lipid-free ApoA-I, recombinant ApoE2, ApoE3, ApoE4	Baseline efflux	None	ABCA1 is involved in mediating cholesterol efflux to ApoA-I and ApoE	[39]
Abca1-deficient mouse primary cultured astrocytes	ApoE	Baseline efflux	None	↓ ApoE and cholesterol in CSF lipoproteins	[41]
Rat astrocytes and human astrocytes	ApoA-I, ApoE, and HDL	ABCA1- and ABCG1-mediated efflux	Ethanol, cAMP, or 22(R)-HC plus 9-cis-RA	↑ ABCA1- and ABCG1-mediated efflux	[143]
Murine neuronal cell line HT-22	HDL alone or HDL associated with ApoE3 or ApoE4	Baseline efflux	None	No differences in cholesterol efflux depending on ApoE isoform	[145]
Abcg1−/− and Abcg4−/− primary astrocytes	HDL	Baseline efflux	None	↓ Efflux levels	[144]
HEK293	HDL	Baseline efflux	Overexpression of Abcg1 and Abcg4 (transfection)	↑ Efflux levels	
Primary neurons and ApoE-deficient astrocytes	Recombinant ApoE3 and ApoE4	Baseline efflux	None	Recombinant ApoE3: ↑ efflux compared to recombinant ApoE4	[147]
Human THP-1 monocytes, J774 macrophages, and SR-BI-enriched Fu5AH cells	Plasma, HDL, and ApoA-I from AD patients (n = 39) and controls (n = 20)	Baseline efflux, SR-BI- or ABCA1-mediated efflux	J774 + cAMP	↓ ABCA1-mediated efflux	[33]
Primary cortical astrocytes and neurons	ApoA-I, HDL, ApoE3	ABCA1-, ABCG1-, and ABCG4-mediated efflux	Ethanol, 22-HC plus 9cis-RA, ABCA1, ABCG1, and ABCG4 siRNAs, probucol (ABCA1 inhibitor)	ABCA1 and ABCG1 are mainly involved in cholesterol efflux in astrocytes, whereas ABCG4 regulates it in neurons	[52]
BHK cells	CSF from AD (n = 26), MCI (n = 35), and control (n = 47) individuals	ABCA1-mediated efflux	Mifepristone (induces ABCA1)	↓ CSF ABCA1-mediated efflux in AD and MCI patients	[149]
J774 macrophages	Plasma-isolated HDL from AD (n = 33), MCI (n = 27), and control (n = 27) individuals	Baseline efflux and ABCA1-mediated efflux	cAMP	↓ HDL baseline efflux in AD patients. No differences in ABCA1-mediated efflux	[30]
ApoE3 and ApoE4 primary astrocytes	BSA	ABCA1-mediated efflux	GW3965 (LXR agonist), CS-6253 (ABCA1 agonist peptide)	↓ ABCA1-mediated efflux from ApoE4 astrocytes	[110]
BHK cells	CSF from non-demented ApoE4/4 (n = 3), ApoE3/4 (n = 9), and non ApoE4 (n = 9) carriers	ABCA1-mediated efflux	Mifepristone	↓ CSF ABCA1-mediated efflux in ApoE4/4 carriers	
J774 macrophages	CSF from AD (n = 37), non-AD dementia patients (n = 16), and controls (n = 39)	Baseline and ABCA1-mediated efflux	cAMP	↓ CSF ABCA1-mediated efflux in AD patients	[150]
CHO-K1 cells		ABCG1-mediated efflux	Expression of hABCG1	↓ CSF ABCG1-mediated efflux in AD patients	
RAW264.7 murine macrophages	HDL isolated from control (n = 24) and AD patient (n = 44) serum	Baseline and ABCA1-mediated efflux	cAMP	↓ HDL ABCA1-mediated efflux in AD patients	[148]
J774 macrophages and microglia cells	Human ApoA-I and HDL. CSF samples (with no demographic or clinical data)	Baseline and ABCA1/G1-mediated efflux	cAMP	Efflux correlates with CSF concentrations of cholesterol, ApoA-I, ApoE, and ApoJ	[151]
A172 astrocytes and SH-SY5Y neurons			T0901317 (LXR agonist)		

ABC: ATP-binding cassette; AD: Alzheimer’s disease; Apo: apolipoprotein; BHK: baby hamster kidney; cAMP: cyclin adenosine monophosphate; CHO: Chinese hamster ovary; CSF: cerebrospinal fluid; HC: hydroxycholesterol; HDL: high-density lipoprotein; LXR: liver X receptor; MCI: mild cognitive impairment; SR-BI: scavenger receptor class B type I.

## Data Availability

Not applicable.

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
