# Peer review of "HDL-like-Mediated Cell Cholesterol Trafficking in the Central Nervous System and Alzheimer’s Disease Pathogenesis"

_ijms, 2022, doi:10.3390/ijms23169356_

Round 1

Reviewer 1 Report

The authors tried to make a Review regarding HDL-like mediated Cell Cholesterol Trafficking in Alzheimer’s Disease. Please see my suggestions regarding this manuscript, which needs an extensive revision:

Pages of the manuscript are not correctly numbered.

Each section must be better developed, as 9 pages of real manuscript (in the MDPI format, where the text occupies 2/3 of a page) cannot be considered a complete Review.

L78-80. Aim of the study is poor and must be better developed. As the topic is not a new one, it is needed of highlighting some aspects by responding to the following questions: Which is the novelty of your study or the special aspects it brings to the field? What makes different your study from others in the same/similar topic, already published? Why have the authors have chosen this topic as the literature is plenty of details on this topic?

I suggest a new Section 2. Literature selection, after accessing WoS and checking the number of the papers in the topic, using the key words the authors provided after the Abstract. As the authors have stated that this is a Review, a PRISMA flow chart is recommended. I suggest checking both Page et al. papers, where this type of graphic is very well described: Page, M.J.; McKenzie, J.E.; Bossuyt, P.M.; Boutron, I.; Hoffmann, T.C.; Mulrow, C.D.; Shamseer, L.; Tetzlaff, J.M.; Akl, E.A.; Brennan, S.E.; et al. The PRISMA 2020 statement: An updated guideline for reporting systematic reviews. Journal of Clinical Epidemiology 2021, 134, 178-189, doi:10.1016/j.jclinepi.2021.03.001. Page, M.J.; McKenzie, J.E.; Bossuyt, P.M.; Boutron, I.; Hoffmann, T.C.; Mulrow, C.D.; Shamseer, L.; Tetzlaff, J.M.; Moher, D. Updating guidance for reporting systematic reviews: development of the PRISMA 2020 statement. Journal of Clinical Epidemiology 2021, 134, 103-112, doi:10.1016/j.jclinepi.2021.02.003.  and will help you to provide a correct PRISMA flow chart. Please take care and detail in the best way the inclusion/exclusion criteria used for the literature selection. Include here also the MeSH terms. Do not forget renumbering the following sections.

Figure 1 is good but blurred. Please provide best quality one. I suggest cropping them directly from the original shape, not saving them in different other formats which lose clarity.

Only 1 figure and 1 Table for a Review type paper?

More idea must be developed. I suggest describing the potential use of statins in the prevention of Alzheimer Disease. Detail if novel medication such as PCSK-9 inhibitors could improve the circulation of lipids in the brain. Also, please develop the potential impact of antioxidant compounds in order to decrease the level of LDL-cholesterol and the alteration of neuron membrane done by free radicals ( I suggest checking very recent and relevant papers ( https://doi.org/10.1007/s12035-020-02065-3https://doi.org/10.3389/fphar.2020.01097 ; https://doi.org/10.1007/s11356-021-17830-7https://doi.org/10.1007/s11064-021-03415-w ; https://doi.org/10.1007/s12035-020-02211-x etc) 

Author Response

RESPONSES TO REVIEWER 1:

We acknowledge the Reviewer for constructive criticism, which significantly helped us to improve the manuscript.

Pages of the manuscript are not correctly numbered.

We have addressed this point.

Each section must be better developed, as 9 pages of real manuscript (in the MDPI format, where the text occupies 2/3 of a page) cannot be considered a complete Review.

Due to the scope of the review, i.e. to describe the mechanisms underlying the HDL-like mediated cholesterol trafficking in the central nervous system (CNS) and its implications in Alzheimer’s Disease (AD) pathogenesis, we comprehensively searched in Pubmed all the manuscripts containing information on HDL-like mediated cholesterol trafficking in the central nervous system, particularly those related with AD (see below for more details). We have now changed the title of the review article to clarify that it is mainly focused on “HDL-like mediated Cell Cholesterol Trafficking in the Central Nervous System and Alzheimer’s Disease Pathogenesis”. Now, this review article contains almost 5400 words, excluding references and abstract (Managing Editor recommended 4000 or more words), a Table that summarizes all data exploring HDL-mediated cholesterol efflux and cholesterol uptake processes regarding CNS/AD in different cell types and one figure that provides, in a comparative layout, the differences between the HDL-like-mediated cholesterol trafficking in a healthy an AD brain.

L78-80. Aim of the study is poor and must be better developed. As the topic is not a new one, it is needed of highlighting some aspects by responding to the following questions: Which is the novelty of your study or the special aspects it brings to the field? What makes different your study from others in the same/similar topic, already published? Why have the authors have chosen this topic as the literature is plenty of details on this topic?

We partially agree with the comment of the Reviewer when stating that “the topic is not new one”. Indeed, much information has been published regarding the cholesterol transport, in general, and apoE in CNS in relation with AD. However, regarding the mechanisms underlying the HDL-like mediated cholesterol trafficking in the central nervous system (CNS) and its implications in Alzheimer’s Disease (AD) pathogenesis, only two review articles have included a similar topic in the last 5 years (one in BBA -without open access- although, more focused, in general, in neurodegenerative disorders and another one in Frontiers in Physiology, although more focused on the exchange of systemic HDL and their implications in neurodegenerative disorders). Therefore, the novelty of this Review is that we critically review all relevant data on HDL-like mediated cholesterol trafficking in the central nervous system, particularly those data that have implications in AD pathogenesis. Overall, evidence seems to point in the direction of decreased cholesterol efflux to CSF’s HDL-like lipoproteins among patients with AD, although most of the studies have been performed in cells that are not physiological relevant for CNS pathology, representing a major flaw in this field. As regards HDL-like mediated cholesterol uptake studies, much less information is available and many questions remain to be answered. Therefore, we strived to build a critical review on these two points, hoping that the present approach would be of interest to broad audiences in lipoproteins and AD disciplines. The fact that more than 25% of the studies cited in the manuscript have been published over the last 5 years supports the view that the individual topics presented are of current interest. These points have now been included in the last paragraph of the introduction and the abstract (highlighted).

I suggest a new Section 2. Literature selection, after accessing WoS and checking the number of the papers in the topic, using the key words the authors provided after the Abstract. As the authors have stated that this is a Review, a PRISMA flow chart is recommended. I suggest checking both Page et al. papers, where this type of graphic is very well described: Page, M.J.; McKenzie, J.E.; Bossuyt, P.M.; Boutron, I.; Hoffmann, T.C.; Mulrow, C.D.; Shamseer, L.; Tetzlaff, J.M.; Akl, E.A.; Brennan, S.E.; et al. The PRISMA 2020 statement: An updated guideline for reporting systematic reviews. Journal of Clinical Epidemiology 2021, 134, 178-189, doi:10.1016/j.jclinepi.2021.03.001. Page, M.J.; McKenzie, J.E.; Bossuyt, P.M.; Boutron, I.; Hoffmann, T.C.; Mulrow, C.D.; Shamseer, L.; Tetzlaff, J.M.; Moher, D. Updating guidance for reporting systematic reviews: development of the PRISMA 2020 statement. Journal of Clinical Epidemiology 2021, 134, 103-112, doi:10.1016/j.jclinepi.2021.02.003.  and will help you to provide a correct PRISMA flow chart. Please take care and detail in the best way the inclusion/exclusion criteria used for the literature selection. Include here also the MeSH terms. Do not forget renumbering the following sections.

For writing this review, PubMed was searched comprehensively with combinations of the keywords AD, HDL and cholesterol efflux and, also, by combining dementia, apoE and cholesterol trafficking. Next, all papers that described findings related with the mechanisms underlying the HDL-like mediated cholesterol trafficking in the CNS and its implications in AD pathogenesis were included in the review article. This point has now been included at the end of introduction. Key words have been changed to follow these criteria.

Figure 1 is good but blurred. Please provide best quality one. I suggest cropping them directly from the original shape, not saving them in different other formats which lose clarity.

As requested, we have added a high-resolution image.

Only 1 figure and 1 Table for a Review type paper?

As commented above, all manuscripts containing information on HDL-like mediated cholesterol trafficking in the CNS are summarized in the Table 1 and discussed in the text. Taken into the account the different pieces of the review manuscript, Figure 1 provides the main differences between the HDL-like-mediated cholesterol trafficking in a healthy and AD brain.

More idea must be developed. I suggest describing the potential use of statins in the prevention of Alzheimer Disease. Detail if novel medication such as PCSK-9 inhibitors could improve the circulation of lipids in the brain. Also, please develop the potential impact of antioxidant compounds in order to decrease the level of LDL-cholesterol and the alteration of neuron membrane done by free radicals ( I suggest checking very recent and relevant papers ( https://doi.org/10.1007/s12035-020-02065-3; https://doi.org/10.3389/fphar.2020.01097 ; https://doi.org/10.1007/s11356-021-17830-7; https://doi.org/10.1007/s11064-021-03415-w ; https://doi.org/10.1007/s12035-020-02211-x etc)

We acknowledge the Reviewer for this important comment. As requested, we have added a new section 3 to describe therapeutic strategies that could potentially impact in CNS cholesterol trafficking and AD. However, in most of these cases, the potential of these strategies to alter HDL-like mediated cholesterol transport in CNS cells remains largely unknown and deserve further investigation.

Reviewer 2 Report

This review Cara Borras et al., reviewed the mechanism of lipoprotein mediated cholesterol trafficking in AD, which authors described in detail the mechanism of transport of cholesterol and cholesterol efflux to HDL-like particles in AD with table of brain cholesterol trafficking studies from several publications. I have minor corrections and suggestion:

In line 47:Involvement of lipids in AD pathology: authors explained the role of lipids in brain and their role in inflammation and brain health but they need to explain in detail the role of lipids other than cholesterol in the pathogenesis of AD and in line 68 what is the role of E4 allele of APOE gene in genetic and pathogenesis of AD?

In line 83:  the SNC is misspelled, it is CNS not SNC

 In line 219; the SNC is misspelled, it is CNS not SNC 

Author Response

RESPONSE TO REVIEWER 2

We acknowledge the Reviewer for his/her positive comments and constructive criticism.

In line 47: Involvement of lipids in AD pathology: authors explained the role of lipids in brain and their role in inflammation and brain health but they need to explain in detail the role of lipids other than cholesterol in the pathogenesis of AD and in line 68 what is the role of E4 allele of APOE gene in genetic and pathogenesis of AD?

As requested, we have summarized evidence that supports the role of other lipid classes in AD pathogenesis (section 1.2) and that the role of APOE4 in AD pathogenesis is discussed in section 2.2. Also, we have included that the role of HDL-like lipoproteins in mediating the trafficking of these lipids in CNS remains largely unknown.

In line 83:  the SNC is misspelled, it is CNS not SNC

In line 219; the SNC is misspelled, it is CNS not SNC

We have corrected these spelling errors.

Reviewer 3 Report

The current study attempts to review the mechanisms via which lipoprotein-mediated cholesterol trafficking through the central nervous system (CNS) occurs in the context of Alzheimer’s disease (AD).  Very clearly are presented during the review the research directions where there is still a need for elucidations for decreased cholesterol efflux to CSF’s HDL-like lipoproteins among patients with AD.

I found the paper to be overall well written and I felt confident that the authors performed careful literature research. The table 1 contains well structured and relevant information regarding previous brain cholesterol trafficking studies.

However, I recommend that a minor revision of the manuscript is warranted. My concerns, I would ask you to consider, is regarding the writing font in figure 1, it is too little and is not visible, please corrected with at least two larger font units.

Author Response

RESPONSE TO REVIEWER 3

We acknowledge the Reviewer for the positive comments on our work and his/her encouraging view on it. We have now included criteria of literature searching in the Introduction and improved the resolution of figure 1.

However, I recommend that a minor revision of the manuscript is warranted. My concerns, I would ask you to consider, is regarding the writing font in figure 1, it is too little and is not visible, please corrected with at least two larger font units.

As requested, we have added a high-resolution image and increased writing font.

Round 2

Reviewer 1 Report

The paper/sections remained poorly developed, no figure/table added, no changes were highlighted in the "revised" manuscript, almost none of my content-related suggestions was followed. Please see my previous Review report and proceed. Half of this manuscript is about references.

Author Response

RESPONSES TO REVIEWER 1:

We acknowledge the Reviewer for constructive criticism, which significantly helped us to improve the manuscript. Please find in the new manuscript all the recommended changes highlighted in red.

Pages of the manuscript are not correctly numbered.

This point has been addressed.

Each section must be better developed, as 9 pages of real manuscript (in the MDPI format, where the text occupies 2/3 of a page) cannot be considered a complete Review.

According to the reviewer’s suggestion, we have developed each section further. We have also included a new section regarding possible therapeutic approaches (new section 3). All the new information included in the text is highlighted in red. 

The review article currently contains over 6000 words (we have extended the last version in 2000 words), excluding references and abstract (Managing Editor recommended 4000 words), 1 figure and 3 tables (2 new ones). The first table (new) includes all the relevant information regarding cholesterol transporters and receptors involved in cholesterol efflux and uptake processes in the brain (including information of brain cell expression, regulation and main functions). The second table (new) includes how the different isoforms of ApoE affect Alzheimer’s disease (AD) in the brain (particularly those processes related with cholesterol trafficking), and the third table summarizes all the data exploring HDL-mediated cholesterol efflux and cholesterol uptake processes regarding CNS/AD in different cell types. Accordingly, the text contains all this new information as well. Regarding the figure provided in the manuscript, we believe that it’s a very detailed schematic representation of cholesterol efflux in the brain, with a comparative layout of the differences between the HDL-like-mediated cholesterol trafficking in a healthy an AD brain. We believe that the manuscript is a review considering the number of words as well as the number of figure/tables and the sections included. In our opinion, it covers all the information related to the main scope of the review. We believe that the text meets now the reviewer´s criteria.

L78-80. Aim of the study is poor and must be better developed. As the topic is not a new one, it is needed of highlighting some aspects by responding to the following questions: Which is the novelty of your study or the special aspects it brings to the field? What makes different your study from others in the same/similar topic, already published? Why have the authors have chosen this topic as the literature is plenty of details on this topic?

We only partially agree with the Reviewer’s comment when stating that “the topic is not a new one”. While we completely agree that a lot of information regarding brain cholesterol transport and ApoE in the CNS in the context of AD has already been published, the number of works or more specific reviews exploring the underlying mechanisms by which HDL-like mediated cholesterol trafficking in the central nervous system (CNS) occurs in AD pathogenesis are very scarce. Two slightly related review articles have been published in the last 5 years (one in Biochimica Biophysica Acta -without open access- more focused in neurodegenerative disorders, and another one in Frontiers in Physiology, more focused on the exchange of systemic HDL and their implications in neurodegenerative disorders). Therefore, the novelty of this Review is that we critically reviewed all relevant data on HDL-like mediated cholesterol trafficking in the central nervous system, particularly those data that have implications in AD pathogenesis. Overall, evidence seems to point in the direction of decreased cholesterol efflux to CSF’s HDL-like lipoproteins among patients with AD, although most of the studies have been performed in cells that are not physiological relevant for CNS pathology, representing a major flaw in the field. As regards HDL-like mediated cholesterol uptake studies, much less information is available and many questions remain to be answered. Therefore, we strived to build a critical review on these two points, hoping that the present approach would be of interest to broad audiences in lipoproteins and AD disciplines. Moreover, and considering that more than 25% of the studies cited in the manuscript have been published over the last 5 years, it seems clear that the view of the topics presented are of current interest.

However, and taking your suggestions into consideration, we believe that the aim of the study needed to be better defined. Therefore, we have tried to further develop your questions regarding novelty of the topic. These points have now been included in the last paragraph of the introduction section and the abstract and have been highlighted in red.

I suggest a new Section 2. Literature selection, after accessing WoS and checking the number of the papers in the topic, using the key words the authors provided after the Abstract. As the authors have stated that this is a Review, a PRISMA flow chart is recommended. I suggest checking both Page et al. papers, where this type of graphic is very well described: Page, M.J.; McKenzie, J.E.; Bossuyt, P.M.; Boutron, I.; Hoffmann, T.C.; Mulrow, C.D.; Shamseer, L.; Tetzlaff, J.M.; Akl, E.A.; Brennan, S.E.; et al. The PRISMA 2020 statement: An updated guideline for reporting systematic reviews. Journal of Clinical Epidemiology 2021, 134, 178-189, doi:10.1016/j.jclinepi.2021.03.001. Page, M.J.; McKenzie, J.E.; Bossuyt, P.M.; Boutron, I.; Hoffmann, T.C.; Mulrow, C.D.; Shamseer, L.; Tetzlaff, J.M.; Moher, D. Updating guidance for reporting systematic reviews: development of the PRISMA 2020 statement. Journal of Clinical Epidemiology 2021, 134, 103-112, doi:10.1016/j.jclinepi.2021.02.003.  and will help you to provide a correct PRISMA flow chart. Please take care and detail in the best way the inclusion/exclusion criteria used for the literature selection. Include here also the MeSH terms. Do not forget renumbering the following sections.

We highly appreciate the reviewer’s input on how to perform a systematic review on terms of literature selection for our manuscript. However, our goal was to perform a literature review rather than a systematic review. According to this definition, a systematic review answers a focused clinical question eliminating bias, whereas a literature review provides a summary or overview of a topic. We believe that the review suited for our goal is the literature review, as our aim was not destined to answer a clinical question. Due to the scope of our review, i.e. a specific question describing the mechanisms underlying the HDL-like mediated cholesterol trafficking in the CNS and its implications in AD pathogenesis, we comprehensively searched Pubmed for all the manuscripts containing information on HDL-like mediated cholesterol trafficking in the CNS, particularly those related with AD (see below for more details). Parallelly, we have changed the title of the review article to clarify that it is mainly focused on “HDL-like mediated Cell Cholesterol Trafficking in the Central Nervous System and Alzheimer’s Disease Pathogenesis”. 

More specifically, and to explain how PubMed was used, we comprehensively searched for combinations of the following keywords: AD, HDL and cholesterol efflux and, also, by combining dementia, ApoE and cholesterol trafficking. Next, all papers that described findings related with the mechanisms underlying the HDL-like mediated cholesterol trafficking in the CNS and its implications in AD pathogenesis were included in the review article. This point has now been included at the end of the introduction section. Key words have also been changed to follow these criteria. All this new information has now been included in the text and highlighted in red.

Figure 1 is good but blurred. Please provide best quality one. I suggest cropping them directly from the original shape, not saving them in different other formats which lose clarity.

As requested, we have added a high-resolution image.

Only 1 figure and 1 Table for a Review type paper?

As commented above, all the reviewed manuscripts involving brain HDL-like-mediated cholesterol trafficking studies in the CNS in the context of AD and their main information are summarized in the old Table 1 (now Table 3) and discussed in the text. The table includes all the works exploring cholesterol efflux and cholesterol uptake processes regarding AD in different cell types ordered chronologically. Taken into the account the different sections of our review, Figure 1 provides the main differences between the HDL-like-mediated cholesterol trafficking in a healthy and AD brain. To our opinion, Figure 1 and Table 3 summarize the most important information of the manuscript. However, and according to your suggestion, 2 new tables have been included: the first one includes all relevant information regarding cholesterol transporters and receptors involved in cholesterol efflux and uptake processes in the brain (including information of brain cell expression, regulation as well as main functions); and the second one includes how the different isoforms of ApoE affect AD in the brain. Accordingly, all this information has been included in the text as well and highlighted in red.

We hope that the performed changes and the two new included tables will meet the reviewer’s suggestions.

More idea must be developed. I suggest describing the potential use of statins in the prevention of Alzheimer Disease. Detail if novel medication such as PCSK-9 inhibitors could improve the circulation of lipids in the brain. Also, please develop the potential impact of antioxidant compounds in order to decrease the level of LDL-cholesterol and the alteration of neuron membrane done by free radicals ( I suggest checking very recent and relevant papers ( https://doi.org/10.1007/s12035-020-02065-3; https://doi.org/10.3389/fphar.2020.01097 ; https://doi.org/10.1007/s11356-021-17830-7; https://doi.org/10.1007/s11064-021-03415-w ; https://doi.org/10.1007/s12035-020-02211-x etc)

We acknowledge the Reviewer for his/her comment. As requested, we have also added a new section 3 to describe therapeutic strategies that could potentially impact in CNS cholesterol trafficking and AD (highlighted in red). However, in most of these cases, the potential of these strategies to alter HDL-like mediated cholesterol transport in CNS cells remains largely unknown and deserve further investigation.

Round 3

Reviewer 1 Report

The paper remained not enough developed for IJMS level. Graphical part is also poor with a single figure. I cannot find any novelty or interesting aspect included in this manuscript.

Author Response

RESPONSES TO REVIEWER 1:

We are sorry to hear about the reviewer’s 1 opinion regarding the quality of our manuscript. We believe that the novelty and interesting aspects of our works rely on the fact that all the works exploring cholesterol efflux and cholesterol uptake processes regarding AD in different cell types were reviewed. According to our knowledge, no similar reviews regarding this topic exist.  However, we hope that the minor changes required by the editor team will make the manuscript more appealing.

RESPONSES TO ACADEMIC EDITOR:

According to your suggestion, the present version of the manuscript now includes a better description of our research strategy based on the PRISMA statement. The following information has been included in the manuscript in a new section of the introduction “1.3. Literature Search Strategy”:

“A literature review was performed based on the “Preferred Reporting Items for Systematic Reviews and MetaAnalyses” (PRISMA) statement. Relevant studies from peer-reviewed journals were identified from three electronic databases (PubMed, Google Scholar and Web of Science) up to June 1st, 2022, without any language restriction. Three groups of medical subject terms were applied, including “Alzheimer disease”, “cholesterol trafficking”, “central nervous system” and “cholesterol efflux”. To identify additional studies and reviews, combinations of specific keywords were also performed: dementia, ApoE and HDL. Hand searching of reference lists in the included reviews was also performed. Three of the authors (C.B., J.C.E-G., and M.T.) independently screened articles, extracted relevant data, and assessed the quality of the studies. For the works exploring cholesterol efflux and cholesterol uptake processes regarding AD in different cell types, a uniform table was prepared to collect related characteristics, including first author, year of publication, cell culture, sample used (cholesterol acceptor), mechanism tested, activation, and main findings. All papers describing findings related with the mechanisms underlying the HDL-like mediated cholesterol trafficking in the CNS and its implications in AD pathogenesis were included, whereas similar papers related to non-AD’s dementia were excluded. The process was agreed upon by all authors.”